# A proteome-wide genetic investigation identifies several SARS-CoV-2-exploited host targets of clinical relevance

Mohd Anisul[1,2]*, Jarrod Shilts[1], Jeremy Schwartzentruber[1,2], James Hayhurst[2,3], Annalisa Buniello[2,3], Elmutaz Shaikho Elhaj Mohammed[4], Jie Zheng[5], Michael Holmes[6,7], David Ochoa[2,3], Miguel Carmona[2,3], Joseph Maranville[4], Tom R Gaunt[5], Valur Emilsson[8,9], Vilmundur Gudnason[8,9], Ellen M McDonagh[2,3], Gavin J Wright[1,10], Maya Ghoussaini[1,2]*†, Ian Dunham[1,2,3]*†

[1]Wellcome Sanger Institute, Wellcome Genome Campus, Cambridge, United Kingdom; [2]Open Targets, Wellcome Genome Campus, Hinxton, United Kingdom; [3]European Molecular Biology Laboratory, European Bioinformatics Institute (EMBL-EBI), Wellcome Genome Campus, Cambridge, United Kingdom; [4]Bristol-Myers Squibb, Cambridge, United States; [5]Medical Research Council (MRC) Integrative Epidemiology Unit, Department of Population Health Sciences, University of Bristol, Bristol, United Kingdom; [6]Clinical Trial Service Unit and Epidemiological Studies Unit (CTSU), Nuffield Department of Population Health, University of Oxford, Oxford, United Kingdom; [7]Medical Research Council Population Health Research Unit (MRC PHRU), Nuffield Department of Population Health, University of Oxford, Oxford, United Kingdom; [8]Icelandic Heart Association, Kopavogur, Iceland; [9]Faculty of Medicine, University of Iceland, Reykjavik, Iceland; [10]Department of Biology, York Biomedical Research Institute, Hull York Medical School, University of York, York, United Kingdom

*For correspondence:
mk31@sanger.ac.uk (MA);
mg29@sanger.ac.uk (MG);
dunham@ebi.ac.uk (ID)

†These authors are Joint Senior Authors to this work

## Abstract

**Background:** The virus SARS-CoV-2 can exploit biological vulnerabilities (e.g. host proteins) in susceptible hosts that predispose to the development of severe COVID-19.

**Methods:** To identify host proteins that may contribute to the risk of severe COVID-19, we undertook proteome-wide genetic colocalisation tests, and polygenic (pan) and cis-Mendelian randomisation analyses leveraging publicly available protein and COVID-19 datasets.

**Results:** Our analytic approach identified several known targets (e.g. ABO, OAS1), but also nominated new proteins such as soluble Fas (colocalisation probability >0.9, p=1 × 10⁻⁴), implicating Fas-mediated apoptosis as a potential target for COVID-19 risk. The polygenic (pan) and cis-Mendelian randomisation analyses showed consistent associations of genetically predicted ABO protein with several COVID-19 phenotypes. The *ABO* signal is highly pleiotropic, and a look-up of proteins associated with the *ABO* signal revealed that the strongest association was with soluble CD209. We demonstrated experimentally that CD209 directly interacts with the spike protein of SARS-CoV-2, suggesting a mechanism that could explain the ABO association with COVID-19.

**Conclusions:** Our work provides a prioritised list of host targets potentially exploited by SARS-CoV-2 and is a precursor for further research on CD209 and FAS as therapeutically tractable targets for COVID-19.

**Funding:** MAK, JSc, JH, AB, DO, MC, EMM, MG, ID were funded by Open Targets. J.Z. and T.R.G were funded by the UK Medical Research Council Integrative Epidemiology Unit (MC_UU_00011/4). JSh and GJW were funded by the Wellcome Trust Grant 206194. This research was funded in part by

the Wellcome Trust [Grant 206194]. For the purpose of open access, the author has applied a CC BY public copyright licence to any Author Accepted Manuscript version arising from this submission.

## Introduction

At the current time, the coronavirus disease 2019 (COVID-19) pandemic is implicated in the deaths of more than 4 million people worldwide (*Dong et al., 2020*). Although effective vaccines have been developed to substantially reduce mortality and morbidity due to severe COVID-19, the emergence of mutated strains of the SARS-CoV-2 virus has challenged the effectiveness of existing vaccines and raised the urgency of identifying alternate therapeutic pathways to target the virus (*Tegally, 2020*; *Erik et al., 2020* ; *Collier et al., 2021*). Nevertheless, it is likely that the mutated strains of SARS-CoV-2 will continue to exploit the same vulnerable host biology to bind onto and infect cells and, in susceptible individuals, evade immune defences and promote the excessive host inflammatory response that is characteristic of severe COVID-19 (*Gordon et al., 2020a*). Therefore, the identification of host proteins that play roles in COVID-19 susceptibility and severity remains crucial to the development of therapeutics as host protein mechanisms are independent of genomic mutations in the virus. An improved understanding of these therapeutically relevant virus-host pathways may also be important in combating viruses beyond SARS-CoV-2 (*Perrin-Cocon et al., 2020*).

Several large-scale systematic experimental efforts have identified key host proteins that interact with viral proteins in the pathogenesis of severe COVID-19 (*Gordon et al., 2020a*; *Gordon et al., 2020b*; *Bouhaddou et al., 2020*). These notably include efforts to identify direct interactions with the spike protein of SARS-CoV-2, which mediates virus attachment onto receptors to infect host cells and is also the basis of most vaccines (*Shang et al., 2020*; *Harvey et al., 2021*). To complement in vitro host protein characterisation efforts, several groups have leveraged genetic datasets of human proteins and COVID-19 disease to identify therapeutically actionable candidate host proteins that are likely to play roles in enhancing COVID-19 susceptibility or to be involved in the pathogenesis of severe COVID-19 (*Pairo-Castineira et al., 2021*; *Zhou et al., 2021*). One of the approaches used was Mendelian randomisation (MR). MR simulates the design of randomised trials, with the underlying principle that randomisation of alleles at conception offers the opportunity to examine approximate differences in average risk of disease between comparable groups in a population that differ only in the distribution of the risk factor of interest (*Davies et al., 2018*), for example, protein abundance (*Zheng et al., 2020*). This allows the use of alleles as genetic instruments representing genetically predicted protein levels to proxy effects of pharmacological modulation of the protein. Some of the clinically actionable proteins identified by the MR approach are part of type I interferon signalling (encoded by genes: *IFNAR2, TYK2, OAS1*) and interleukin-6 (IL-6) signalling pathways (*IL6R*). Only one of these proteins (encoded by *OAS1*) had any evidence of genetic colocalisation, that is, evidence that genetic associations of the protein and COVID outcomes shared the same causal genetic signal (*Zhou et al., 2021*). An additional protein that was supported by both MR and genetic colocalisation tests was ABO (*Zhou et al., 2021*), reported in several published genome-wide association studies (GWAS) of COVID-19 (*Pairo-Castineira et al., 2021*; *Ellinghaus et al., 2020*). In response to the first published GWAS of COVID-19, we reported findings that link the *ABO* signal with a number of clinically actionable targets including coagulation factors (von Willebrand factor [vWF], and Factor VIII [F8]), IL-6, and CD209/DC-SIGN (*Karim et al., 2020*).

However, in most of the previous MR studies (*Pairo-Castineira et al., 2021*; *Zhou et al., 2021*), investigators only used curated cis-acting variants (genetic variants near or in the gene encoding the relevant protein) as genetic instruments to represent effects of genetically predicted protein concentrations, rather than genome-wide instruments. While the use of cis-acting variants can minimise the risk of horizontal pleiotropic effects (i.e. associations driven by other proteins not on the causal pathway for the disease), it can suffer from lower power than a genome-wide analysis due to fewer available instruments (*Zheng et al., 2020*). Furthermore, in previous protein-COVID-19 MR studies, genetic colocalisation tests were carried out only for protein-phenotype associations that were significant in the MR analysis, potentially excluding many protein-phenotype associations that may share the same causal genetic signal but are underpowered in a proteome-wide MR approach.

In the present study, we expanded on these previous reports by undertaking a proteome-wide two-sample pan- and cis-MR analysis using the Sun et al. GWAS (*Sun et al., 2018*) of plasma protein

**eLife digest** Individuals who become infected with the virus that causes COVID-19 can experience a wide variety of symptoms. These can range from no symptoms or minor symptoms to severe illness and death. Key demographic factors, such as age, gender and race, are known to affect how susceptible an individual is to infection. However, molecular factors, such as unique gene mutations and gene expression levels can also have a major impact on patient responses by affecting the levels of proteins in the body. Proteins that are too abundant or too scarce may mean the difference between dying from or surviving COVID-19.

Identifying the molecular factors in a host that affect how viruses can infect individuals, evade immune defences or trigger severe illness, could provide new ways to treat patients with COVID-19. Such factors are likely to remain constant, even when the virus mutates into new strains. Hence, insights would likely apply across all virus strains, including current strains, such as alpha and delta, and any new strains that may emerge in the future.

Using such a 'natural experiment' approach, Karim et al. compared the genetic profiles of over 30,000 COVID-19 patients and a million healthy individuals. Nine proteins were found to have an impact on COVID-19 infection and disease severity. Four proteins were ranked as top priorities for potential treatment targets. One protein, called CD209 (also known as DC-SIGN), is involved in how the virus enters the host cells, and had one of the strongest associations with COVID-19. Two proteins, called IL-6R and FAS, were involved in the immune response and could be responsible for the immune over-activation often seen in severe COVID-19. Finally, one protein, called OAS1, formed part of the body's innate antiviral defence system and appeared to reduce susceptibility to COVID-19.

Knowing more about the proteins that influence the severity of COVID-19 opens up new ways to predict, protect and treat patients who may have severe or fatal reactions to infection. Indeed, one of the identified proteins (IL-6R) had already been targeted in recent clinical trials with some encouraging results. Considering CD209 as a potential receptor for the virus could provide another avenue for therapeutics, similar to previously successful approaches to block the virus' known interaction with a receptor protein. Ultimately, this research could supply an entirely new set of treatment options to help combat the COVID-19 pandemic.

concentrations and several COVID-19 GWAS phenotypes from the ICDA COVID-19 Host Genetics Initiative (October 2020 release) (*Huang et al., 2020*). First, we showed that genetically predicted circulating ABO protein was associated with COVID-19 susceptibility and severity and the lead *ABO* signal was associated strongly with plasma concentrations of soluble CD209. Second, we collected evidence for a direct mechanism of interaction between the SARS-CoV-2 spike protein and human CD209 protein. Third, we performed proteome-wide genetic colocalisation tests, followed by single-instrument cis-MR analysis, and we report additional novel targets of therapeutic relevance. Finally, we examined associated phenotypes using the colocalising signals from the Open Targets Genetics portal (http://genetics.opentargets.org) to shed light on the biological basis of association of the proteins with the COVID-19 phenotypes.

## Materials and methods

**Key resources table**

| Reagent type (species) or resource | Designation | Source or reference | Identifiers | Additional information |
|---|---|---|---|---|
| Cell line (*Homo sapiens*) | HEK293-E | Yves Durocher, PMID:11788735 | RRID:CVCL_6974 | |
| Transfected construct (*Homo sapiens*) | pCMV6-CD209 | Origene | Cat.# SC304915 | Plasmid for CD209 cDNA expression in cell-based binding assay |
| Transfected construct (*Homo sapiens*) | pTT3-ACE2-BLH | PMID:33432067 | | Plasmid for recombinant ACE2 extracellular domain, for plate-based assays as the immobilised form |

*Continued on next page*

*Continued*

| Reagent type (species) or resource | Designation | Source or reference | Identifiers | Additional information |
|---|---|---|---|---|
| Transfected construct (*Homo sapiens*) | pTT3-CD209-BLH | This paper | | Plasmid for recombinant CD209 extracellular domain for plate-based assays as the immobilised form |
| Transfected construct (*Homo sapiens*) | pTT3-Cd4d3+ d4 | Addgene | RRID:Addgene_32402 | Plasmid for recombinant tag control (Cd4 domains 3 and 4) |
| Transfected construct (*Homo sapiens*) | pTT3-SPIKE-COMP-BLac | This paper | | Plasmid for recombinant SARS-CoV-2 spike extracellular domain for plate-based assays as the soluble form |
| Transfected construct (*Homo sapiens*) | pTT3-BirA-FLAG | Addgene | RRID:Addgene_64395 | Biotin ligase plasmid for recombinant protein biotinylation |
| Peptide, recombinant protein | Streptavidin R-phycoerythrin | BioLegend | Cat.# 405245 | For tetramer staining in cell-based binding assay |
| Chemical compound, drug | DAPI (4',6-diamidino-2-phenylindole) | BioLegend | Cat.# 422801 | 1 μM for flow cytometry live/dead staining |
| Chemical compound, drug | D-biotin | Sigma-Aldrich | Cat.# 2031 | 100 μM supplemented to cell culture media for biotinylation |
| Software, algorithm | R (version 4.0.3) | R Foundation | www.r-project.org RRID:SCR_001905 | Analysis and generating plots |

## Genetic associations of proteins

We primarily used Sun et al. protein GWAS data (*Sun et al., 2018*; *Emilsson et al., 2018*) for the pan-/cis-MR analyses and for performing genetic colocalisation tests (described below). The pan-/cis-MR effects were expressed per standard deviation (SD) higher genetically predicted plasma protein concentrations. Two additional proteomic datasets (*Emilsson et al., 2018*; *Suhre et al., 2017*) were used to identify proteins associated with the *ABO* locus. The genotyping protocols and QC of these proteomic studies have been described previously (*Sun et al., 2018*; *Emilsson et al., 2018*; *Suhre et al., 2017*). All three of the proteomic studies have used the SOMAscan assay platform (an aptamer-based protein detection platform) to detect and quantify protein abundance (*Gold et al., 2012*).

## Genetic associations of COVID-19

We used seven meta-analysed COVID-19 datasets from the October 2020 release of the ICDA COVID-HGI group (https://www.covid19hg.org/results/r4/). These seven COVID-19 outcomes are A1 (very severe respiratory confirmed COVID vs. not hospitalised COVID), A2 (very severe respiratory confirmed COVID vs. population), B1 (hospitalised COVID vs. not hospitalised COVID), B2 (hospitalised COVID vs. population), C1 (COVID vs. lab/self-reported negative), C2 (COVID vs. population), and D1 (predicted COVID from self-reported symptoms vs. predicted or self-reported non-COVID). Definitions of these outcomes are provided in *Supplementary file 1*.

## Harmonisation of protein and COVID summary statistics

Prior to analyses, we performed a liftover of datasets that reported genomic coordinates using the GRCh37 assembly to GRCh38. We also checked and ensured that the effect allele in a GWAS locus is the alternative allele in the forward strand of the reference genome. To infer strand for palindromic variants (variants with A/T or G/C alleles, i.e. variants with the same pair of letters on the forward strand as on the reverse strand), we first checked the orientation of all non-palindromic variants with respect to the reference genome to assess whether there was a strand consensus of 99% or more. For example, for a given GWAS, if ≥99 % of the non-palindromic variants were on the forward strand, we assumed that the palindromic variant would also be on the forward strand; otherwise, they were excluded from analyses. Details of the harmonisation workflow are provided in our GitHub pages (*EBISPOT, 2020*; *Opentargets Inc, 2021*).

## Mendelian randomisation

To construct genetic instruments for MR analysis, we selected near-independent ($r^2 = 0.05$) genetic variants from across the genome ('*pan*'-instruments) or from within ±1 Mbp from the transcription

start site (TSS) of the gene encoding the protein ('*cis*'-instruments) associated with the encoded protein abundance at p≤5 × $10^{-8}$ for pan-MR analyses and at a less stringent p ≤ 1 × $10^{-5}$ for cis-MR analyses (this p-value corrects for the number of proteins in the druggable genome *Schmidt, 2020*). We used the generalised summary data-based Mendelian randomisation (GSMR) approach with the heterogeneity-independent instrument (HEIDI)-outlier flag turned on to carry out the pan- and cis-MR analyses (*Zhu et al., 2018*). The GSMR software, using the HEIDI-outlier method, removes potentially pleiotropic instruments and accounts for the residual correlation between instruments (important as we are using near-independent genetic instruments). To select near-independent genetic instruments and account for linkage disequilibrium (LD) in the MR analyses, we used genotype data from 10,000 randomly sampled UK Biobank participants to create a reference LD matrix, which is ancestry-matched to the pQTL data we used. For each COVID-19 outcome, we used the Benjamini–Hochberg FDR (False Discovery Rate) threshold of 5 % for significance, adjusting for 2042 tests in cis-MR analyses and 1286 tests in pan-MR analyses. For trans-acting instruments in pan-MR associations, variants were mapped to their respective cis-gene that had the highest overall V2G score in the Open Targets Genetics portal (*Ghoussaini, 2021*; *Mountjoy, 2020*; *Open Targets Genetics, 2019a*).

## Colocalisation analysis and phenome-wide association study

To identify shared causal genetic signals between protein and COVID outcomes, we used the Bayesian method of genetic colocalisation implemented in the *coloc* R package (*Giambartolomei, 2014*) using the marginal association statistics for each trait (i.e. assuming one independent signal in each region). We used beta and standard errors of cis-pQTLs of phenotype pairs as inputs. The default priors in *coloc* were used, that is, the prior of an SNP (single nucleotide polymorphism)-trait association is 1 × $10^{-4}$, and the prior of an SNP associating with both traits is 1 × $10^{-5}$. For each COVID-19 outcome, a posterior probability for shared causal genetic signal (PP.H4) threshold of more than 0.8 was used to identify shared causal genetic variants. For colocalising signals, we carried out a phenome-wide association study (PheWAS) using GWAS summary statistics (n = ~ 3000 GWAS) from the Open Targets Genetics portal (*Ghoussaini, 2021*; *Mountjoy, 2020*).

## Evidence against aptamer binding artefacts

For variants associated with proteins due to aptamer or epitope binding artefacts (which tend to be missense variants) (*Joshi and Mayr, 2018*), we first assessed whether genetic instruments for MR or coloc-based single-SNP MR analysis were associated with corresponding gene expression (i.e. whether they were also cis-eQTLs). This used gene expression data from the Open Targets Genetics portal (*Ghoussaini, 2021*). SNPs that were not cis-eQTLs were investigated further by identifying whether they were (or were in LD at $r^2 = 0.8$ with) missense variants. To query if variants were missense or in LD with missense variants, we used the functional consequence data from Open Targets Genetics (*Ghoussaini, 2021*) (which used gnomAD v2 for variant effect prediction annotation, *Lek, 2016*). The reasoning was, if missense variants also had effects on corresponding gene expression, the causal inference using the missense variants as genetic instruments was unlikely to be biased even if the effect estimates were invalid.

Where cis-pQTLs were not cis-eQTLs and were missense variants (or in LD with missense variants at $r^2 = 0.8$) affecting the respective genes, these proteins were flagged and excluded from any further downstream analyses on the basis that the missense variant(s) might influence aptamer binding and produce biased effect estimates. Where cis-pQTLs were also cis-eQTLs and were missense variants (or in LD with missense variants) for the respective genes, although the effect estimates would not be valid, the causal inference using the instruments is unlikely to be biased; hence, these variants were retained in supplementary files and estimates of probes represented by these variants were flagged (using an asterisk) in the main figures. The rest, where cis-pQTLs had an effect on gene expression but were not missense variants or in LD with missense variants, were included in all analyses and presented without restrictions.

## Recombinant protein production

Recombinant human receptors and SARS-CoV-2 spike protein extracellular domains were expressed and purified as previously described (*Shilts et al., 2021*). Briefly, the full extracellular domain sequences of each were expressed as soluble secreted proteins in HEK293 cells. All proteins were

affinity-purified using their hexahistidine tags. For biotinylated proteins, co-transfection of secreted BirA ligase in the presence of 100 μM D-biotin resulted in the covalent addition of a biotin group to an acceptor peptide tag, also as described previously (*Kerr and Wright, 2012*). The extracellular domain of CD209 (Q9NNX6) was defined as beginning at Pro114, while the full cDNA sequence was acquired from OriGene (#SC304915).

## Plate-based protein binding assay

The binding of biotinylated human receptor extracellular domains to pentameric SARS-CoV-2 spike protein was measured using the avidity-based extracellular interaction assay (AVEXIS) as previously described (*Bushell et al., 2008*). Briefly, the wells of a streptavidin-coated 96-well plate were saturated with biotinylated bait of either CD209, ACE2, or a previously described negative-control construct consisting only of the C-terminal protein tags shared by all other recombinant proteins (rat Cd4(d3 +4)-linker-Bio-6xHis) (*Voulgaraki, 2005*; *Galaway and Wright, 2020*). Across these baits, we applied a dilution series of the full SARS-CoV-2 spike protein extracellular domain pentamerised by a peptide sequence from the cartilage oligomeric matrix protein with a beta lactamase reporter. After washing, binding was measured by hydrolysis of a colorimetric nitrocefin substrate whose product was quantified by light absorbance at 450 nm.

## Cell-based receptor binding assay

HEK293 cells were transiently transfected as described previously (*Bartholdson, 2012*) with expression plasmids encoding full-length cDNA of CD209 (Origene SC304915), or a mock transfection lacking the expression plasmid. Separately, recombinant biotinylated spike protein was tetramerised around streptavidin conjugated to phycoerythrin as previously described (*Sharma et al., 2018*). Cells were incubated with tetramers of spike or a control construct of protein tags before being analysed on a flow cytometer as previously described (*Shilts et al., 2021*). HEK293 cell lines were provided by Yves Durcohcer (National Research Council, Canada). Cell lines were authenticated upon first receipt by DNA sequencing. All cell lines were regularly tested for mycoplasma by PCR (Surrey Diagnostics) and found to be negative all throughout experiments. These cell lines are not listed by ICLAC as 'commonly misidentified'.

## Code availability

Codes used to harmonise summary statistics are provided in https://github.com/EBISPOT/gwas-sumstats-harmoniser (*EBISPOT, 2020*). Codes for pan- and cis-MR analyses are provided on the GSMR website (https://cnsgenomics.com/software/gcta/#GSMR). Codes for genetic colocalisation analyses are provided on the coloc GitHub page (https://github.com/chr1swallace/coloc; *Wallace, 2021*). All codes used in the paper to reproduce results are provided in https://github.com/mohdkarim/covid_paper (copy archived at swh:1:rev:4ab9f9b17ffde57f7831ea555394290ba240a2b9; *Anisul, 2021*).

## Results

### Pan- and cis-MR analyses support the role of circulating ABO protein concentrations and soluble IL-6R in COVID-19 risk

Our multi-instrument MR analysis used both genetic variants from across the genome (pan-MR) and genetic variants near or in the gene encoding the relevant protein (cis-MR) to investigate associations of genetically predicted plasma protein concentrations with the risk of COVID-19 outcomes. The COVID-19 outcome definitions are provided in *Supplementary file 1*. Although the pan-MR analysis leveraged genetic data from both cis- and trans-acting pQTLs (with a selection of pQTLs from across the genome automated by GSMR's built-in HEIDI-outlier exclusion method), for some protein-COVID-19 pairs that were associated at 5 % FDR, the associations with COVID-19 outcomes were exclusively driven by trans-acting pQTLs or cis-acting genetic instruments. For example, although six proteins were represented by both cis- and trans-acting genetic instruments, two (ABO and IL6R) were represented only by cis-acting variants and one (SELE) was driven entirely by trans-acting instruments (mainly *ABO* trans-pQTLs) (*Supplementary file 2*). Overall, the pan-MR analysis revealed nine distinct protein probes associated with four COVID outcomes at an FDR of 5 % (*Figure 1A*). The pQTLs selected by GSMR to represent these nine probes were also cis-eQTLs (as curated for the Open Targets

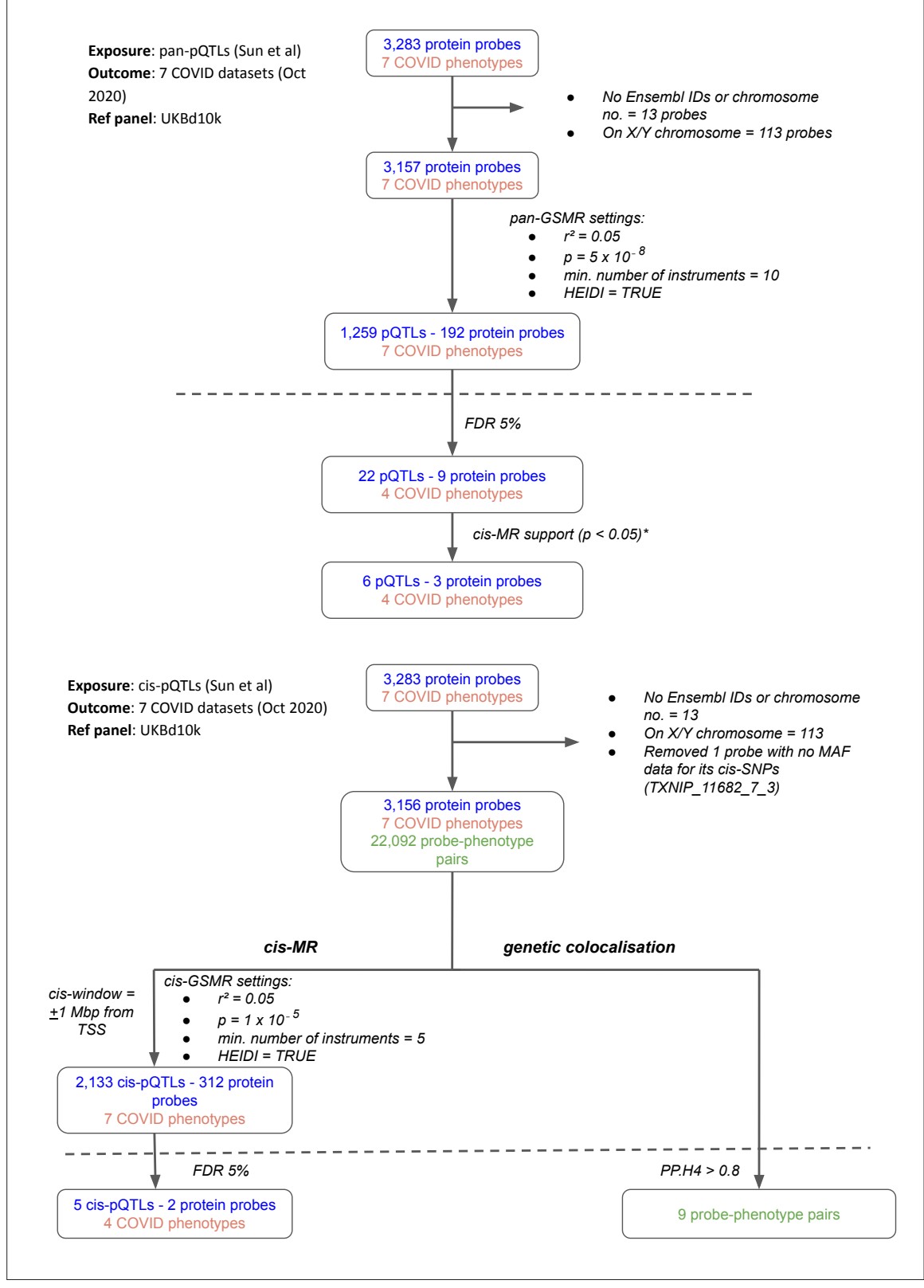

**Figure 1.** Flowcharts illustrating the process of (**A**) pan-Mendelian randomisation (MR) and (**B**) cis-MR and genetic colocalisation. Both pan- and cis-MR methods used (***Sun et al., 2018***) as the source of genetic instruments and the UK Biobank downsampled 10 k (UKBd10k) individual genotype data as reference panel. We selected near-independent genetic instruments and performed two sample MR analysis using generalised summary data-based Mendelian randomisation that adjusted for residual correlation between instruments. Genetic colocalisation analysis was used to estimate posterior

*Figure 1 continued on next page*

Figure 1 continued

probabilities of shared causal genetic signal between protein and outcomes. A posterior probability of shared causal genetic signal of more than 0.6 (i.e. a PP.H4 or posterior probability for hypothesis 4 > 0.6) was used as evidence of genetic colocalisation. The dashed line separates analysis (above the line) from target curation (below the line). *Only three proteins with pan-MR evidence of association with COVID also had cis-MR evidence support at nominal cis-MR p-value<0.05.

Genetics portal *Open Targets Genetics, 2019*) and, except the ABO signal via rs8176719-insertionC (will be referred to as rs8176719-insC – a frameshift mutation that inserts a guanine nucleotide in the 258th position of exon 6), were not missense variants or in LD with missense variants (*Supplementary file 3*), minimising the possibility that SNPs with artefactual associations with proteins were used as genetic instruments for the majority of significant pan-MR association.

While the pan-MR analysis used genetic data from across the genome, the cis-MR analysis restricted genetic instrument selection to those near (within 1 Mb of TSS) or in the gene encoding the protein. Three proteins with pan-MR associations were supported by corresponding cis-MR associations (*Figure 1A and B*, *Supplementary file 4*): ABO, ICAM-1, and IL-6R. Among these three, only ABO and IL-6R proteins had some evidence of genetic colocalisation with posterior probabilities (PP. H4) more than 0.9 and 0.4, respectively, of a shared genetic signal between protein and COVID-19

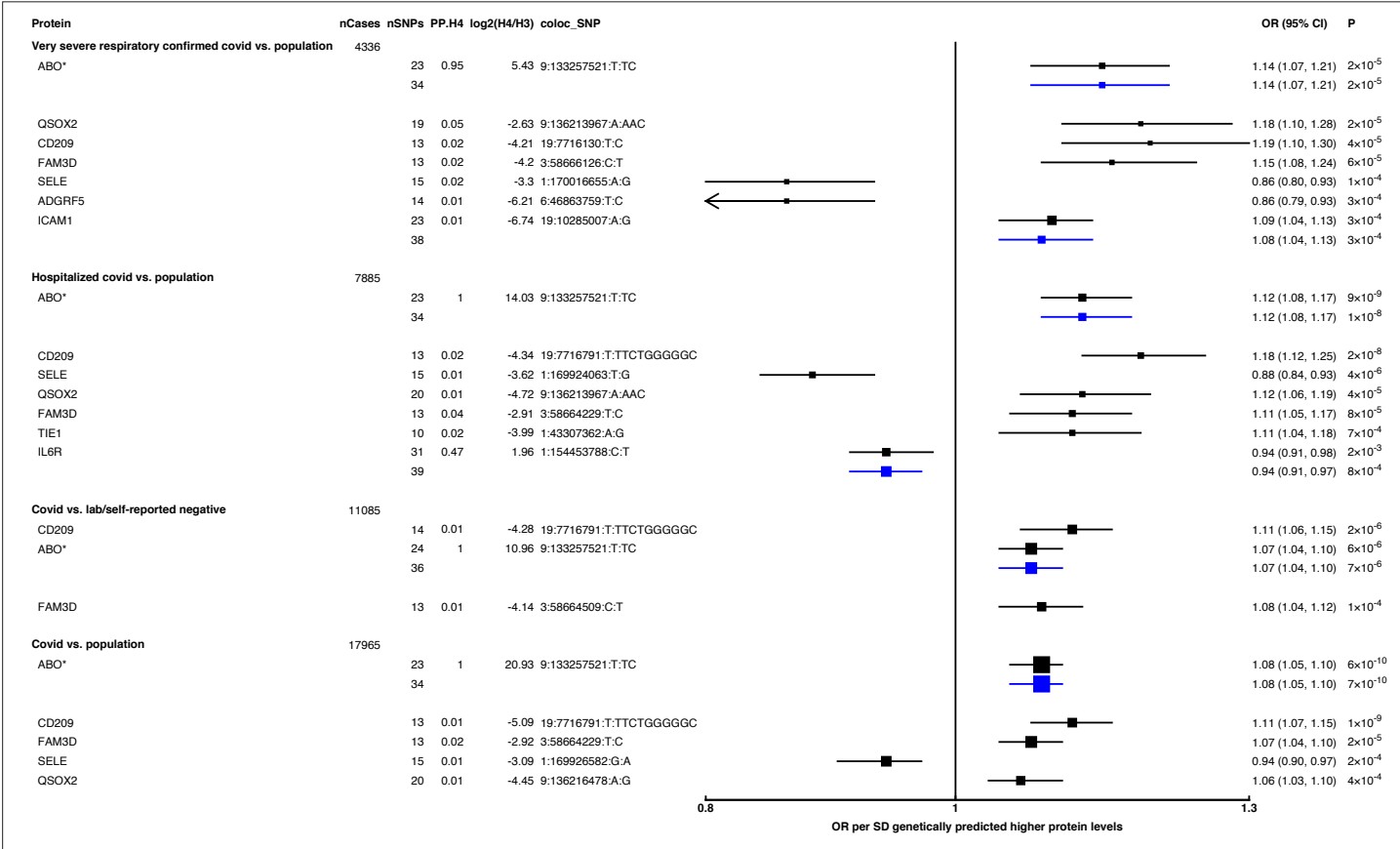

**Figure 2.** Forest plot illustrating associations of genetically predicted plasma protein concentrations with selected COVID-19 phenotypes. The black point estimates represent odds ratios (ORs) of COVID-19 outcome per standard deviation (SD) increase of genetically predicted protein abundance using genetic instruments from across the genome (pan-Mendelian randomisation [pan-MR]). The blue point estimates represent OR of COVID outcome per SD increase of genetically predicted protein abundance using genetic instruments near or in the gene encoding the protein (cis-MR). Error bars represent 95 % confidence intervals (95% CI). The areas of the squares are proportional to the inverse of the variance of the log ORs. For each COVID phenotype, pan-MR associations at FDR 5 % were retained. Each row under a COVID phenotype represents a pQTL and includes the number of cases in the COVID phenotype (nCases), the number of SNPs used as genetic instruments for the protein (nSNPs), the posterior probability that protein and COVID traits colocalise (PP.H4), the posterior probability evidence for vs. against shared causal variants (log2(H4/H3)), and the candidate colocalising signal (coloc_SNP). * denotes proteins that have coloc_SNP that are either missense variants or in linkage disequilibrium with missense variants, rendering their effect estimates potentially biased.

**Table 1.** Summary of proteins reported in our study and the different sources of evidence supporting their prioritisation.

| | Supported by multi-instrument pan-MR | | | Supported by multi-instrument cis-MR | Supported by GC and single-SNP cis-MR | Experimental support | Existing drugs | Previously reported |
|---|---|---|---|---|---|---|---|---|
| Protein | No. of cis-acting SNPs | No. of trans-acting SNPs | Trans-acting gene(s)* | | | | | |
| ABO | 93 | 0 | None | ✓ | ✓ | x | x | ✓ |
| QSOX2 | 16 | 63 | *ABO, OBP2B, ADAMTS13* | x | x | x | x | x |
| CD209 | 8 | 45 | *ABO, SURF6* | x | x | ✓ | x | ✓ |
| FAM3D | 8 | 44 | *ABO, SULT2B1, FAM83E, NTN5, FUT2* | x | x | x | x | x |
| SELE | 0 | 60 | *ABO, FAM118B, RALGDS, OBP2B, ADAMTS13, SURF1* | x | x | x | ✓ | x |
| ADGRF5 | 5 | 9 | *ABO, IL6ST, ADAMTS13* | x | x | x | x | x |
| ICAM1 | 71 | 1 | *ABO* | ✓ | x | x | ✓ | x |
| TIE1 | 2 | 18 | *ABO, ST3GAL6, GBGT1, SURF6* | x | x | x | x | x |
| IL6R | 62 | 0 | None | ✓ | x | x | ✓ | ✓ |
| FAS | No | No | No | x | ✓ | x | x | x |
| OAS1 | No | No | No | x | ✓ | x | x | ✓ |
| THBS3 | No | No | No | x | ✓ | x | x | x |

Detailed description of each column is provided in **Supplementary file 9**.

*Where trans-acting SNPs are used, genes assigned to SNPs with the highest variant-to-gene scores in Open Targets Genetics were used for annotation. GC: genetic colocalisation; MR: Mendelian randomisation.

phenotype (*Figure 2*). Although the PP.H4 of IL-6R was very weak (0.4), it had a positive (H4/H3 = 3.6), indicating a common signal of the IL-6R protein with the COVID-19 outcome is a more likely scenario than the association driven by two independent signals.

Genetically predicted ABO concentration was associated with risk in four out of seven COVID-19 outcomes (*Figure 2*). These four outcomes represented both susceptibility (e.g. COVID-19 vs. population, cis-MR odds ratio [OR] [95% CI] per SD genetically predicted ABO concentrations: 1.08 [1.05, 1.10], p=7 $\times 10^{-10}$) and severity (e.g. hospitalised COVID-19 vs. population, cis-MR OR [95% CI]: 1.12 [1.08, 1.17], p=1 $\times 10^{-8}$) of COVID-19. Genetically predicted soluble IL-6R was only associated with higher risk of hospitalised COVID-19 compared to population-based controls (cis-MR OR [95% CI] per SD genetically predicted IL-6R: 0.94 [0.91, 0.97], p=8 $\times 10^{-4}$) (*Figure 2*).

When examining the SNPs involved in the pan-MR associations of the nine probes, all probes except IL-6R and ABO had at least one trans-acting SNP, and in all these cases, at least one of the trans-acting SNPs were assigned to the *ABO* gene by the Open Targets Genetics V2G pipeline (*Table 1*), re-confirming the pervasive pleiotropy of the *ABO* genetic signal. Furthermore, when examining the consistency of pan-MR associations of these nine probes across all seven COVID-19 outcomes, the protein probes that have trans-acting *ABO* SNPs exhibited a similar association profile as the ABO protein probe, associated with only COVID-19 outcomes that have population-based controls (*Supplementary file 5*).

## CD209/DC-SIGN: a proposed alternate receptor for SARS-CoV-2

The *ABO* signal (rs8176719-insC) contributes to the determination of non-O blood groups and regulates circulating levels of both ABO and several non-ABO proteins; *Yamagata University Genomic Cohort Consortium (YUGCC), 2014*; *Arguinano et al., 2018*. We explored proteome-wide associations of rs8176719-insC in three separate proteomic datasets; *Emilsson, 2020*. Aside from the ABO protein, the *ABO* signal rs8176719-insC showed the strongest association (Sun et al: p=6.03 $\times 10^{-258}$, Emilsson et al: p=1.00 $\times 10^{-307}$, Suhre et al: p=1.27 $\times 10^{-75}$) with higher plasma concentrations

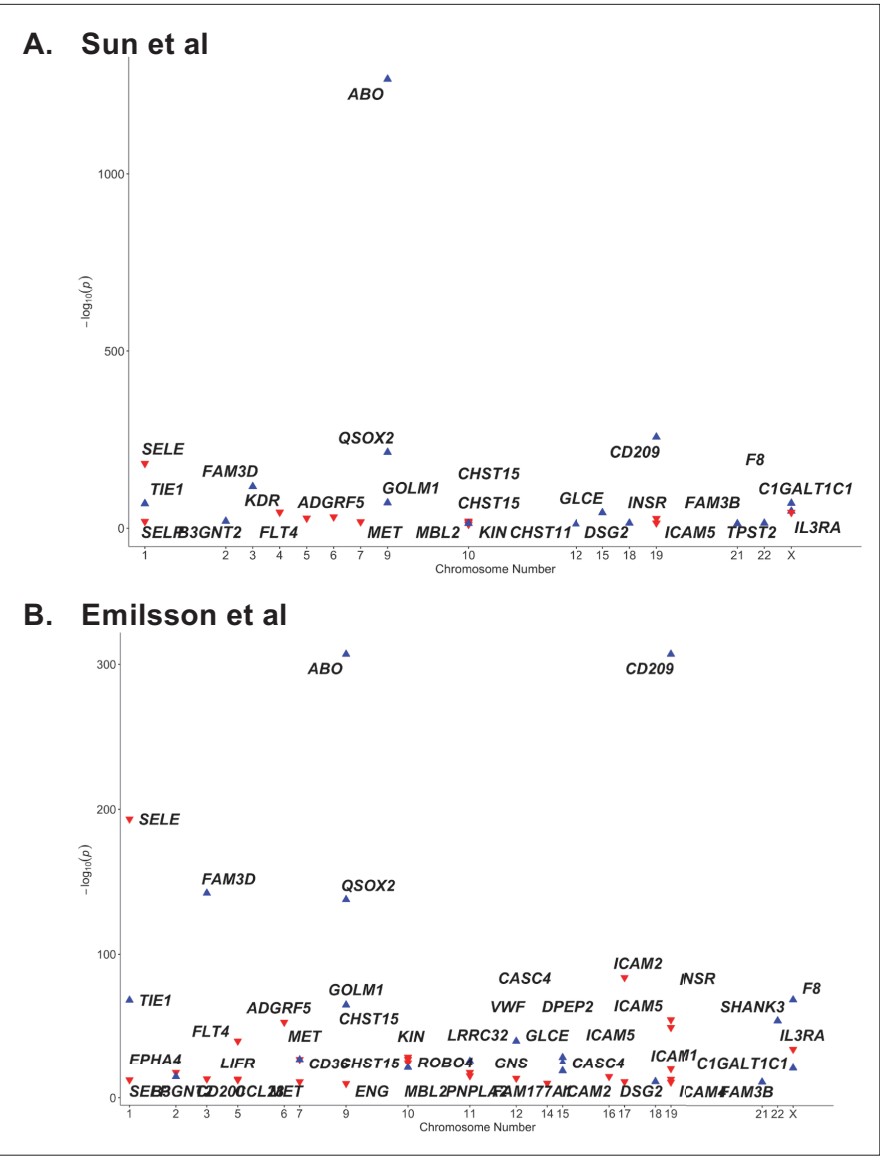

**Figure 3.** Proteome-wide association of the ABO signal (rs8176719-insC) in (**A**) Sun et al. and (**B**) Emilsson et al. datasets. The x-axis represents the chromosome for the gene encoding the protein. The y-axis represents the p-value of the per-allele association of rs8176719-insC (or an SNP in high linkage disequilibrium at $r^2$ >0.8 with rs8176719-insC) with the proteins in Sun et al. and Emilsson et al. datasets. The red triangles point downwards and denote the inverse association of the *ABO* signal with the protein. The blue triangles point upwards and denote the positive association of the *ABO* signal with the protein. Only proteins that were considered significant at the study-specific Bonferroni-corrected p-value thresholds are displayed in this plot and tabulated in ***Supplementary file 6***. (***Supplementary file 6*** also reports associations from an additional protein dataset – Suhre et al.).

of soluble CD209 in all three datasets (associations from two datasets illustrated in ***Figure 3***, and associations from all three datasets tabulated in ***Supplementary file 6***). To validate this as a relevant target for COVID-19, we experimentally tested whether CD209 directly interacts with SARS-CoV-2, as had been recently proposed based on similarities to SARS-CoV-1, which was reported to bind CD209 (***Yang, 2020***). We used human cells to generate recombinant SARS-CoV-2 spike protein, spanning the full-length extracellular domain according to a design previously established to retain functionality (***Shilts et al., 2021***). We found that purified spike protein indeed could directly attach onto human cells expressing CD209 but not control cells, suggesting that CD209 could act as a receptor for viral attachment onto host cells (***Figure 4A***). Furthermore in a direct binding assay testing purified soluble

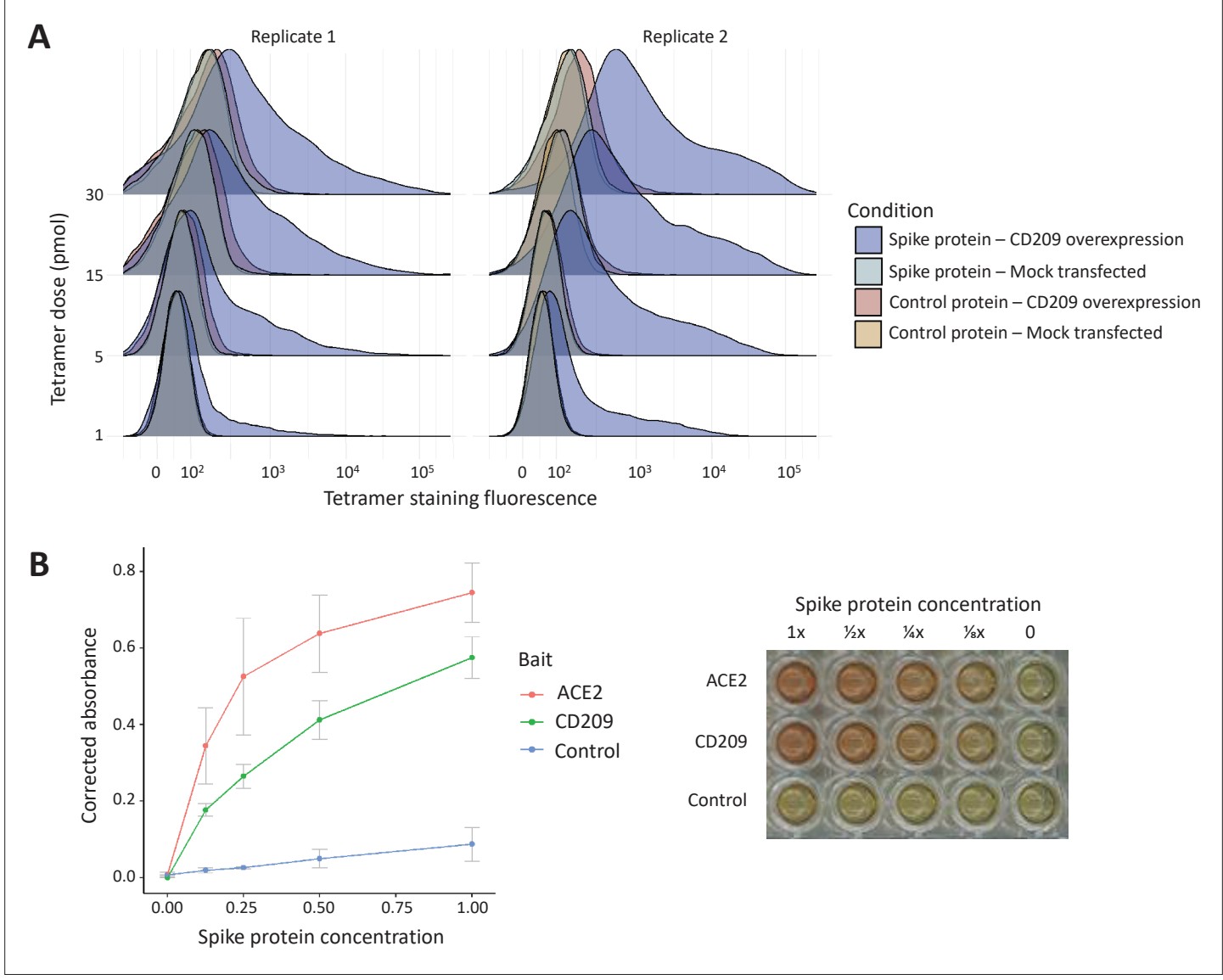

**Figure 4.** In vitro binding experiments with purified SARS-CoV-2 spike protein confirm human CD209 as a functional binding target. (**A**) Human cell lines overexpressing cell-surface CD209 protein gain the ability to specifically bind SARS-CoV-2 spike. The density plots represent flow cytometry measurements of HEK293 cells stained with fluorescently conjugated tetramers of SARS-CoV-2 spike protein or a tag-only protein control. Blue distributions are cells with surface CD209, while red are control-transfected cells. Light shades indicate a negative control tetramer that was used for staining, while dark shades are stained with spike protein. (**B**) Purified recombinant CD209 ectodomains interact with the spike protein of SARS-CoV-2 in an in vitro binding assay. A dilution series of purified spike protein was applied over immobilised CD209, ACE2 (positive control), or a negative control protein. A plot of quantified absorbance is displayed alongside a representative assay plate. Error bars are standard deviations of two replicates.

CD209 and the viral spike protein, we could detect binding that was specific and comparable to the primary known receptor for SARS-CoV-2, ACE2 (*Figure 4B*).

## Proteome-wide genetic colocalisation implicates additional proteins in COVID-19 risk including FAS, SCARA5, and OAS1

To identify additional proteins associated with the risk of COVID-19, we conducted proteome-wide genetic colocalisation tests followed by single-SNP MR analysis (*Supplementary file 7*). This 'coloc-first' approach identified four proteins (ABO, FAS, OAS1, THBS3) with evidence of genetic colocalisation (PP.H4 >0.8) with four out of seven COVID-19 phenotypes (*Figure 5*). Two of these (FAS and THBS3) are, to the best of our knowledge, not reported in proteomic MR studies of COVID-19 to date which have only examined for colocalisation evidence after MR.

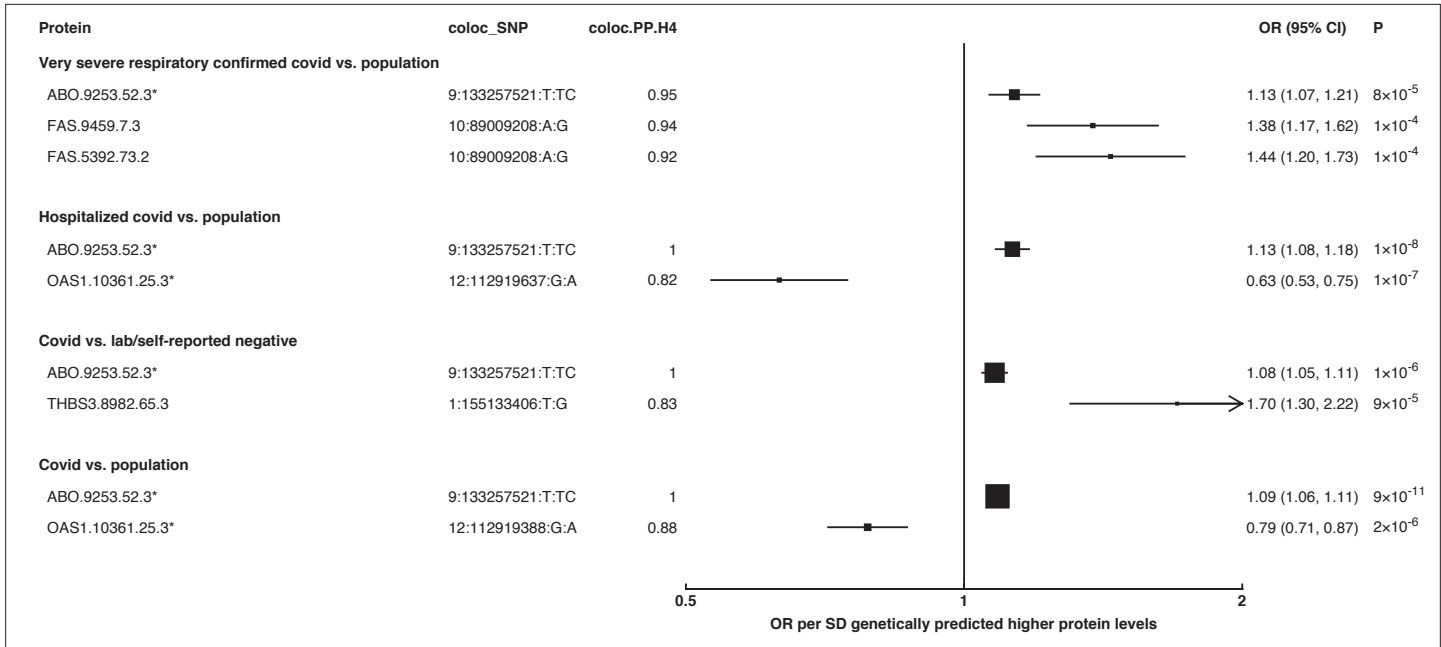

| Protein | coloc_SNP | coloc.PP.H4 | | OR (95% CI) | P |
|---|---|---|---|---|---|
| **Very severe respiratory confirmed covid vs. population** | | | | | |
| ABO.9253.52.3* | 9:133257521:T:TC | 0.95 | | 1.13 (1.07, 1.21) | $8{\times}10^{-5}$ |
| FAS.9459.7.3 | 10:89009208:A:G | 0.94 | | 1.38 (1.17, 1.62) | $1{\times}10^{-4}$ |
| FAS.5392.73.2 | 10:89009208:A:G | 0.92 | | 1.44 (1.20, 1.73) | $1{\times}10^{-4}$ |
| **Hospitalized covid vs. population** | | | | | |
| ABO.9253.52.3* | 9:133257521:T:TC | 1 | | 1.13 (1.08, 1.18) | $1{\times}10^{-8}$ |
| OAS1.10361.25.3* | 12:112919637:G:A | 0.82 | | 0.63 (0.53, 0.75) | $1{\times}10^{-7}$ |
| **Covid vs. lab/self-reported negative** | | | | | |
| ABO.9253.52.3* | 9:133257521:T:TC | 1 | | 1.08 (1.05, 1.11) | $1{\times}10^{-6}$ |
| THBS3.8982.65.3 | 1:155133406:T:G | 0.83 | | 1.70 (1.30, 2.22) | $9{\times}10^{-5}$ |
| **Covid vs. population** | | | | | |
| ABO.9253.52.3* | 9:133257521:T:TC | 1 | | 1.09 (1.06, 1.11) | $9{\times}10^{-11}$ |
| OAS1.10361.25.3* | 12:112919388:G:A | 0.88 | | 0.79 (0.71, 0.87) | $2{\times}10^{-6}$ |

OR per SD genetically predicted higher protein levels

**Figure 5.** Forest plot illustrating associations of genetically predicted plasma protein concentrations that colocalised with the selected COVID-19 phenotypes (PP.H4 > 0.6). The black point estimates represent odds ratios (ORs) of COVID-19 outcome per standard deviation (SD) increase of genetically predicted protein abundance using single-SNP colocalising signals (coloc_SNP). Error bars represent the 95 % confidence interval around the estimates. The areas of the squares are proportional to the inverse of the variance of the log ORs. * denotes proteins that have coloc_SNP that are either missense variants or in linkage disequilibrium with missense variants, rendering their effect estimates potentially biased.

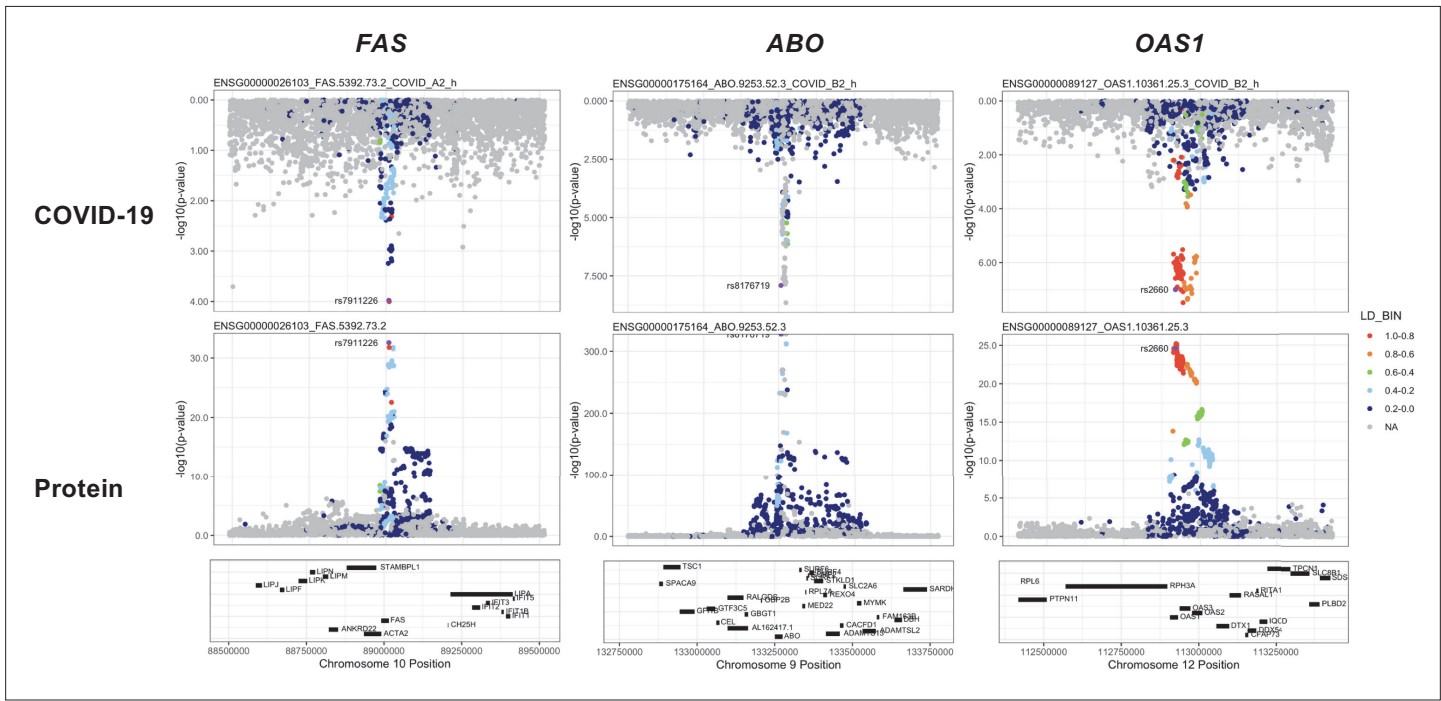

**Figure 6.** Regional association plots arranged to mirror the genetic associations of the colocalising proteins (FAS, ABO, and OAS1) with their respective COVID-19 phenotypes. The top panels represent genetic associations of the selected COVID-19 phenotypes, and the bottom panels represent genetic associations of the protein from the Sun et al. dataset. The x-axis in each panel represents the genomic locations in or around the genes encoding FAS, ABO, and OAS1. The y-axis in each panel represents the p-value of the genetic associations.

Consistent with pan- and cis-MR findings, there was evidence of genetic colocalisation between the ABO protein and six out of seven COVID-19 phenotypes (*Figure 6*), with similar MR estimates when the colocalising SNP was used to perform single-SNP cis-MR.

The coloc-first approach revealed a common genetic signal between OAS1 and COVID-19 in two out of seven COVID-19 phenotypes (PP.H4 = 0.88 in COVID-19 vs. population, and 0.82 in hospitalised COVID-19 vs. population). However, the SNPs representing the common genetic signal between OAS1 and COVID-19 phenotypes (12:112919637:G:A and 12:112919388:G:A) were missense variants in *OAS1* gene or in LD with missense variants at $r^2$ >0.8, rendering their effect estimates potentially biased due to aptamer binding effects (see Materials and methods). Despite this, the same variants have effects on gene expression (as assessed in any of several tissues curated by Open Targets Genetics *Open Targets Genetics, 2019*), which is independent of aptamer binding, suggesting that causal inference regarding OAS1 protein and COVID-19 risk may still be valid. We have, therefore, presented OAS1 estimates in *Figure 5* but flagged with an asterisk denoting the effect estimates as potentially biased.

Unlike ABO, OAS1 could not be tested using the multi-instrument MR approach due to insufficient number of valid instruments, highlighting the complementary value of the genetic colocalisation approach alongside multi-SNP MR methods. In single-SNP MR analyses, genetically predicted higher OAS1 was associated with lower risk of severe COVID-19 vs. population (OR [95% CI] per SD genetically predicted OAS1 concentrations: 0.52 [0.42, 0.65], p=5 × 10$^{-9}$), hospitalised COVID-19 vs. population (0.63 [0.53, 0.75], p=1 × 10$^{-7}$) and susceptibility to COVID-19 vs. population (0.79 [0.71, 0.87], p=2 × 10$^{-6}$).

Proteins that exhibited association only in one of the COVID-19 phenotypes included circulating FAS and THBS3. Genetically predicted elevated FAS (indicated by two FAS probes: FAS.9459.7.3 and FAS.5392.73.2) and THBS3 were associated with a higher risk of severe COVID-19 (OR [95% CI] per SD genetically predicted FAS concentrations indicated by FAS.9459.7.3: 1.38 [1.17, 1.62], p=1 × 10$^{-4}$) and COVID-19 vs. lab/self-reported negative COVID-19 (OR [95% CI] per SD genetically predicted THBS3 concentrations: 1.70 [1.30, 2.22], p=9 × 10$^{-5}$), respectively.

## PheWAS with colocalising variants provides additional biological insights for the basis of associations of the proteins with risk of COVID-19

For the proteins with evidence of genetic colocalisation between protein and COVID-19 phenotype, we used their lead variants (or variants they tag at $r^2$ >0.6 if a lead variant was not reported in a GWAS) to identify additional associated phenotypes. At p<1 × 10$^{-5}$ (Bonferroni corrected for the ~3000 phenotypes in the Open Targets Genetics portal), most of the variants exhibited associations with haematological indices, with some, like the *ABO* signal, also associated with other COVID-19-relevant phenotypes (*Supplementary file 8*). For example, the *ABO* signal was associated with monocyte count, deep vein thrombosis (DVT), and pulmonary embolism (PE). OAS1 and THBS3 variants were associated with platelet counts. For FAS, there were no additional phenotypic associations at p<1 × 10$^{-5}$ shown by its colocalising variant.

## Discussion

Our systematic proteome-wide MR and genetic colocalisation analysis supported several previously proposed proteins and suggested additional clinically actionable targets for COVID-19 (*Table 1*). Of particular note, we provided pan- and cis-MR evidence with strong genetic colocalisation support for the *ABO* signal for most COVID-19 phenotypes. Although the ABO protein itself is not clinically actionable, the *ABO* signal was linked to plasma concentrations of several clinically tractable targets. We demonstrated that the CD209 protein we had found to have the strongest association with this *ABO* signal has a direct interaction with the SARS-CoV-2 spike protein, providing further evidence for a plausible mechanism. Our analyses also supported the role of soluble IL-6R in hospitalised COVID-19, with evidence from pan- and cis-MR analyses but limited evidence of genetic colocalisation with hospitalised COVID-19 but supported by the recent COVID-19 clinical trials of tocilizumab (which is partially mimicked by the IL-6R instrument used in the present study). Using a proteome-wide 'colocalisation-first' approach, we recapitulated previously reported targets (e.g. OAS1) and

uncovered additional novel proteins that may play causal roles in COVID-19 susceptibility (THBS3), or severity (FAS).

Our proteome-wide genetic colocalisation analysis prioritised soluble Fas (sFas, also known as soluble CD95) receptor protein in the very severe COVID-19 phenotype. This finding was not reported in previous proteomic MR studies of COVID-19 most likely because they only assessed evidence of shared signals for targets prioritised by an MR-first approach. The soluble Fas receptor is reported to act as a decoy receptor competing with the trans-membrane Fas receptor for Fas ligand (FasL) (*Cheng, 1994*). Genetically predicted higher circulating sFas is, therefore, likely to represent effects of lower Fas-FasL signalling and, in our study, was associated with a higher risk of very severe COVID-19. Fas-FasL signalling typically initiates a cascade of intracellular programmes that result in cell death or apoptosis. Fas-mediated apoptosis plays a central role in T- and B-cell homeostasis (*Hao, 2008*), preventing the emergence of autoreactive or overactive immune cells (*Hao, 2008*; *Butt, 2015*). Excessive inflammation by hyperactive T-cells and autoantibodies was reported to underlie several cases of severe COVID-19 (*Khamsi, 2021*). To what extent sFas contributes to the excessive pro-inflammatory response in severe COVID-19 remains to be determined. Furthermore, Fas-mediated apoptosis of virus-infected cells is a major mechanism of resolution of viral infections (*Thomson, 2001*). Delayed apoptosis is reported to be one of the strategies exploited by SARS-CoV-2 in the early stages of infection to facilitate viral replication (*Thomson, 2001*; *Ivanisenko et al., 2020*). Additional insights for the role of sFas in COVID-19 can be gleaned from the results of recent drug trials. For example, Fas is one of the major targets of lopinavir-ritonavir – a combination HIV protease inhibitor (*Sorbera et al., 2020*); clinical trials of lopinavir-ritonavir failed to provide any therapeutic benefits beyond standard care in hospitalised COVID-19 patients (*Cao et al., 2020*). On the other hand, clinical trials testing dexamethasone demonstrated beneficial effects on survival for COVID-19 patients who were on respiratory support (*RECOVERY Collaborative Group et al., 2021*). In addition to its anti-inflammatory effects, dexamethasone downregulates molecules associated with decelerating apoptosis (*Achuthan, 2018*), including sFas (*Joashi et al., 2002*). An in-depth assessment of the specific role of soluble Fas in COVID-19, including whether or not it contributes to the beneficial effects of dexamethasone, is warranted in future studies.

Observational studies were the first to report differences in risk of severe COVID-19 based on ABO blood groups, although with some conflicting reports (*Zhao, 2020*; *Zietz et al., 2020*; *Bhattacharjee et al., 2020*). GWAS of COVID-19 susceptibility have, however, consistently reported a signal in the *ABO* locus (*Bhattacharjee et al., 2020*; *Shelton, 2020*), despite prior observations that controls used in the first published GWAS of COVID-19 (*Ellinghaus et al., 2020*) may be over-represented for blood group O (the most common blood group) and can result in associations due to selection bias. However, in a meta-analysis of GWAS of COVID-19, the *ABO* signal remained even when Ellinghaus et al. was excluded.

Furthermore, using these GWAS data, we and Katz et al. (in a preprint) (*Katz et al., 2020*; *Karim et al., 2020*) had previously linked the *ABO* signal with CD209/DC-SIGN protein, clotting factors, coagulation disorders, and concentrations of IL-6, all potential risk factors for COVID-19. In the present study, we build on previous work and show consistent cis- and pan-MR associations of genetically predicted circulating ABO protein with an expanded list of COVID phenotypes which colocalise with the *ABO* signal, supporting a shared genetic signal of ABO protein and the COVID-19 phenotypes.

We show that, next to the ABO protein, the *ABO* signal had the strongest association with the CD209 protein relative to other proteins and present experimental evidence of binding of CD209 with the full-length spike protein of SARS-CoV-2, independently but consistent with a concurrent preprint (*Amraie, 2020*). CD209 is a receptor on monocyte-derived dendritic cells (moDCs) that was shown, before this binding interaction was known, to facilitate entry of replication-competent SARS-CoV-2 and demonstrated to switch off the type I interferon signalling pathways necessary for transcription of several antiviral genes (*Yang, 2020*). The soluble isoforms of CD209 measured in our proteome datasets are known to correlate in expression levels to the membrane isoforms (*Mummidi et al., 2001*; *Plazolles, 2011*), making it plausible that the signal we observed is associated with greater abundance of CD209 as a cell-surface viral receptor. Alternatively, in the context of other viruses known to directly bind CD209 as we show here for SARS-CoV-2, soluble CD209 has been demonstrated to modulate infection, such as by promoting endocytosis if the soluble CD209 coating the virus acts as opsonins (*Plazolles, 2011*). Further research would be beneficial to reveal which of these mechanisms

explain the association we observed. These findings may also help interpret the clinical significance of the higher CD209 gene expression in immune cells (extracted from bronchoalveolar lavage fluid) in severe COVID patients than healthy controls (*Gao, 2020*). It should be noted that the present study developed a therapeutic hypothesis for CD209 based solely on the strong evidence of association of the *ABO* signal with plasma concentrations of CD209 and evidence from the pan-MR association of CD209 with COVID-19 phenotypes (the pan-MR associations being driven mainly by trans-acting *ABO* SNPs) with no corresponding support of cis-MR or colocalisation. This suggests that while cis-MR and colocalisation analyses can support pan-MR associations of a target with disease, the lack of cis-MR or colocalisation for a target is not necessarily evidence against its therapeutic relevance.

In the present study, we also found that genetically predicted higher OAS1 – an interferon-induced broad-spectrum antiviral enzyme – was associated with lower risk of both susceptibility and severity of COVID-19, consistent with findings of a recent published report (*Zhou et al., 2021*). A large clinical trial of systemically administered interferons failed to show any substantial therapeutic benefits for severe COVID-19 (*WHO Solidarity Trial Consortium et al., 2021*). However, the strong evidence from human genetics supports reconsidering the role of interferon-based therapies in a new light, especially with respect to timing of administration (which current genetic studies are unable to provide any insights on) and route (systemic vs. nebulised) (*Monk, 2020*).

Non-O blood group individuals generally have higher risk of DVT and other coagulation disorders than O blood group individuals (*Groot, 2020*). The *ABO* signal, which largely determines the non-O blood groups, was also associated with DVT, PE, and higher levels of vWF and F8; vWF binds to and protects F8 from biological degradation (*Federici, 2003*). F8 is a key protein in the intrinsic coagulation pathway that activates Factor X and induces the formation of fibrin – the central component of blood clots (*Bhopale and Nanda, 2003*). Both DVT and PE are reported to affect almost a third of ICU-admitted COVID-19 patients (*Malas, 2020*). While several clinical trials evaluating the efficacy of anticoagulants for severe COVID-19 are underway, the National Institute of Clinical Excellence in the UK has suggested screening all hospitalised COVID-19 patients for any contraindications to anticoagulant use and offering prophylactic anticoagulation to eligible patients (*National Institute for Health and Care Excellence, 2020*).

We found moderate evidence for the role of IL-6 signalling in COVID-19 in agreement with a previous report (*Bovijn et al., 2021*). However, there was ambiguous evidence of genetic colocalisation (PP.H4: 0.46). Nevertheless, there was more support for a shared genetic signal between sIL-6R and hospitalised COVID-19 than for them to be driven by independent signals (H4/ H3 = 3.6). As noted by Bovign and colleagues (*Bovijn et al., 2021*), with some caveats, the phenotypic consistency of associations between the IL-6R genetic instrument and pharmacological effect of tocilizumab enable potential use of the IL-6R instrument to investigate therapeutic or adverse effects of tocilizumab. Although a previous report showed largely neutral effects of tocilizumab compared to placebo in hospitalised COVID-19 patients (*Stone et al., 2020*), two recent trials (REMAP-CAP *Anthony C and Paul R, 2021* and RECOVERY *RECOVERY Collaborative Group, 2021*) with a longer follow-up period showed beneficial effects on survival at 90 days, consistent with the prediction of a protective effect using the tocilizumab-mimicking IL-6R genetic instrument in the present study and the previous report.

The major strengths of our study include the use of both genome-wide and local genetic instruments for MR analysis, the proteome-wide genetic colocalisation tests to nominate additional proteins of therapeutic relevance, and the expanded list of COVID-19 phenotypes analysed. We showed consistency of the association of ABO with the different COVID-19 phenotypes for both instrument selection strategies. Proteome-wide colocalisation tests implicated additional proteins that likely lacked sufficient genetic instruments to be detected by the multi-instrument GSMR method. For our top-ranked association with the CD209 protein, we provide experimental evidence for a mechanism that implicates CD209 as having a potentially causal role in disease pathology. Our experiments provide both direct evidence of biochemical binding between the purified spike protein of SARS-CoV-2 and CD209, and verification that this interaction occurs in live human cells. Host-directed therapies involving pathogen binding receptors have previously been developed against other infectious diseases where pathogen mutations or variants stymied more traditional approaches (*Zenonos et al., 2015*).

Our study also has several limitations. The reliability of the MR approach depends on the selection of the appropriate genetic instruments for the exposure (*Schmidt, 2020*). Where proteins are the

exposure, the use of genetic instruments from across the genome can result in more instruments and potentially higher power to detect associations. However, the inclusion of a broader set of genetic instruments for protein-MR analysis can lead to associations not mediated by the protein under investigation (i.e. horizontal pleiotropy). In these cases, the use of genetic variants near or in the locus encoding the protein (cis-acting SNPs) can provide more specific estimates of risk, albeit at a potential power cost, associated with genetically predicted concentrations of the protein under investigation (*Schmidt, 2020*). A key problem of the latter approach is the selection of correlated genetic instruments that can lead to numerical approximation errors (*Gkatzionis et al., 2021*). In the present study, we leveraged both pan- and cis-MR approaches and used an MR method (GSMR) that automates the selection of near-independent genetic instruments and performs MR adjusting for any residual correlation (*Zhu et al., 2018*). Nevertheless, horizontal pleiotropy can also affect cis-MR analyses when different variants from the same gene region represent different biological pathways, indicated by heterogeneous effect estimates, or driven by a single variant with a large effect (e.g. missense variants) (*Gkatzionis et al., 2021*). To prevent the selection of heterogeneous instruments and minimise the selection of variants with large effects, the multi-instrument GSMR method used in the present study implements the HEIDI test which excludes genetic variants with strong or heterogeneous effects. The exclusion of missense variants with potential aptamer binding effects is evidenced in our study, where SNPs in 96 % of nominally significant protein probes associated with COVID-19 also had effects on corresponding gene expression in different tissues across gene expression datasets as curated by our portal (*Open Targets Genetics, 2019*). Even while using cis-acting genetic instruments, the MR associations can be confounded due to LD between cis-pQTLs and disease-associated SNPs, and this is at least partially mitigated by genetic colocalisation tests. However, the genetic colocalisation tests used in our study assumed a single causal variant in each locus and will, therefore, result in higher false-negative tests if there is more than one trait-associated causal variant. An additional issue is related to the selection of COVID-19 GWAS datasets used for analyses. Most protein-MR studies have used COVID-19 phenotypes with population-based controls, given their larger number of controls providing additional power to detect signals but at a cost of not being able to distinguish signals relevant to disease progression. While study designs with milder/asymptomatic cases as controls are useful to study disease progression, they are frequently underpowered and, because the selection of study participants are conditioned on the outcome, are susceptible to collider-stratification bias (*Griffith, 2020*). To enable a comprehensive assessment, we used all published COVID-19 phenotypes (October 2020 freeze), irrespective of controls used and, as expected, found most signals in COVID-19 phenotypes with population-based controls. For one of the targets (CD209), although we experimentally demonstrate binding of CD209 with spike protein of SARS-CoV-2, understanding the functional significance CD209 has on viral entry and any immunological relevance during infection requires further research. Finally, although we nominate several targets that may be therapeutically relevant for COVID-19, clinical trials are required for definitive assessments and to guide therapy. For example, the findings related to the *ABO* signal strongly implicated the adverse role of dysregulated coagulation in COVID-19 specifically in non-O blood group individuals; whether pre-emptive use of anticoagulants guided by blood groups can prevent severe COVID-19 is subject to findings of trials such as the ongoing ACTIV-4 trial (NCT04505774) (*U.S. National Library of Medicine, 2020*).

In conclusion, we integrated genetic investigation with functional assessments of CD209, a receptor in moDCs, and postulated that this target may convey the COVID-19 risk of the *ABO* signal. Based on proteome-wide genetic colocalisation and MR, we also prioritised sFas for more detailed investigations of its therapeutic relevance to severe COVID-19 risk.

## Acknowledgements

MAK, JSc, JH, AB, DO, MC, EMM, MG, and ID were funded by Open Targets. JZ and TRG were funded by the UK Medical Research Council Integrative Epidemiology Unit (MC_UU_00011/4). JSh and GJW were funded by the Wellcome Trust Grant 206194. This research was funded in part by the Wellcome Trust (grant 206194). For the purpose of open access, the author has applied a CC BY public copyright licence to any Author Accepted Manuscript version arising from this submission.

## Additional information

### Competing interests

Mohd Anisul, Jeremy Schwartzentruber, James Hayhurst, Annalisa Buniello, David Ochoa, Miguel Carmona, Ellen M McDonagh, Maya Ghoussaini: Open Targets is a pre-competitive partnership currently involving the Wellcome Sanger Institute, EMBL-EBI, BMS, GSK, and Sanofi. Research is funded by financial and in-kind contributions from each of the partners.. Elmutaz Shaikho Elhaj Mohammed: Open Targets is a pre-competitive partnership currently involving the Wellcome Sanger Institute, EMBL-EBI, BMS, GSK, and Sanofi. Research is funded by financial and in-kind contributions from each of the partners. ES is also a full-time employee of Bristol-Myers Squibb.. Michael Holmes: Dr Holmes has consulted for Boehringer Ingelheim, and in adherence to the University of Oxford's Clinical Trial Service Unit & Epidemiological Studies Unit (CSTU) staff policy, did not accept personal honoraria or other payments from pharmaceutical companies.. Joseph Maranville: JM is a full-time employee of Bristol-Myers Squibb and retains stock or stock options in Bristol-Myers Squibb. The author has no other competing interests to declare.. Tom R Gaunt: TG received grants from Biogen and GlaxoSmithKline. The author has no other competing interests to declare.. Ian Dunham: Open Targets is a pre-competitive partnership currently involving the Wellcome Sanger Institute, EMBL-EBI, BMS, GSK, and Sanofi. Research is funded by financial and in-kind contributions from each of the partners. ID also received travel costs within the last 36 months from Takeda for speaking at their Reverse Translation Symposium. The author has no other competing interests to declare.. The other authors declare that no competing interests exist.

### Funding

| Funder | Grant reference number | Author |
|---|---|---|
| Wellcome Trust | Grant 206194 | Mohd Anisul<br>Jarrod Shilts<br>Jeremy Schwartzentruber<br>Gavin J Wright<br>Maya Ghoussaini |
| Medical Research Council | MC_UU_00011/4 | Jie Zheng<br>Tom R Gaunt |
| Open Target | | Mohd Anisul<br>Jeremy Schwartzentruber<br>James Hayhurst<br>Annalisa Buniello<br>David Ochoa<br>Miguel Carmona<br>Ellen M McDonagh<br>Maya Ghoussaini<br>Ian Dunham |

The funders had no role in study design, data collection and interpretation, or the decision to submit the work for publication.

### Author contributions

Mohd Anisul, Conceptualization, Data curation, Formal analysis, Visualization, Writing – original draft, Writing – review and editing; Jarrod Shilts, Conceptualization, Validation, Writing – original draft, Writing – review and editing; Jeremy Schwartzentruber, Joseph Maranville, Maya Ghoussaini, Conceptualization, Supervision, Writing – review and editing; James Hayhurst, David Ochoa, Data curation, Resources, Software, Writing – review and editing; Annalisa Buniello, Data curation, Resources, Writing – review and editing; Elmutaz Shaikho Elhaj Mohammed, Gavin J Wright, Conceptualization, Investigation, Supervision, Writing – review and editing; Jie Zheng, Tom R Gaunt, Supervision, Writing – review and editing; Michael Holmes, Conceptualization, Supervision, Validation, Writing – original draft, Writing – review and editing; Miguel Carmona, Data curation, Project administration, Resources; Valur Emilsson, Project administration, Resources, Writing – review and editing; Vilmundur Gudnason, Ellen M McDonagh, Resources, Supervision, Writing – review and editing; Ian Dunham, Conceptualization, Project administration, Supervision, Writing – review and editing

## Author ORCIDs
Mohd Anisul (iD) http://orcid.org/0000-0003-2960-6017
Jie Zheng (iD) http://orcid.org/0000-0002-6623-6839
Tom R Gaunt (iD) http://orcid.org/0000-0003-0924-3247
Vilmundur Gudnason (iD) http://orcid.org/0000-0001-5696-0084
Gavin J Wright (iD) http://orcid.org/0000-0003-0537-0863
Ian Dunham (iD) http://orcid.org/0000-0003-2525-5598

## Ethics

Human subjects: All institutions contributing cohorts to the COVID-19 Host Genetics Initiative and INTERVAL (Sun et al) study for proteomics received ethics approval from their respective research ethics review boards. All participants in the INTERVAL study provided informed consent before joining the INTERVAL study with approval from the National Research Ethics (11/EE/0538). Ethics statements of studies that contributed participant data to the COVID-19 Host Genetics Initiative are provided in Supplementary Table 1 of their recently published paper (https://www.nature.com/articles/s41586-021-03767-x).

## Decision letter and Author response

Decision letter https://doi.org/10.7554/eLife.69719.sa1
Author response https://doi.org/10.7554/eLife.69719.sa2

# Additional files

## Supplementary files

• Supplementary file 1. Definitions of COVID outcomes.

• Supplementary file 2. Summary of proteins prioritised by pan-Mendelian randomisation.

• Supplementary file 3. Pan-Mendelian randomisation outcomes at p<0.05, each association divided into cis- or trans-pQTLs.

• Supplementary file 4. Cis-Mendelian randomisation outcomes at p<0.05.

• Supplementary file 5. Evaluation of pan-Mendelian randomisation association of protein probes that have passed the 5 % FDR.

• Supplementary file 6. Protein-wide association studies (at study-specific Bonferroni thresholds) of the *ABO* signal using three proteomic datasets (Sun et al., Emilsson et al., Suhre et al.).

• Supplementary file 7. Proteome-wide genetic colocalisation results.

• Supplementary file 8. Phenome-wide association study (p<0.05) from Open Targets Genetics portal for each colocalising variant.

• Supplementary file 9. Key to *Table 1*.

• Transparent reporting form

## Data availability

Summary data used for genetic analyses are publicly available (Sun et al can be downloaded from GWAS catalog https://www.ebi.ac.uk/gwas/downloads/summary-statistics and COVID-19 HGI summary statistics can be downloaded from their website https://www.covid19hg.org/results/). Data generated from our study are provided in the supplementary files (pan-MR and cis-MR association results filtered at p < 0.05 and no filters applied to colocalisation results).

The following dataset was generated:

| Author(s) | Year | Dataset title | Dataset URL | Database and Identifier |
|---|---|---|---|---|
| Sun BB | 2018 | Genomic atlas of the human plasma proteome | https://www.ebi.ac.uk/gwas/downloads/summary-statistics | GWAS Catalog, GCST005806 |
| COVID-19 Host Genetics Initiative | 2021 | Mapping the human genetic architecture of COVID-19 by worldwide meta-analysis | https://www.nature.com/articles/s41586-021-03767-x#Sec24 | COVID GWAS meta-analysis results, release 4 |

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
