## [Decision Letter]

**Acceptance summary:**

This paper takes advantage of publicly available genomic and proteomic data to identify host proteins that may be exploited by SARS-CoV-2 and involved in regulation of COVID-19 susceptibility and severity. These studies provide a foundation for examining the potential of these host proteins to serve as therapeutic targets.

**Decision letter after peer review:**

Thank you for submitting your article "A proteome-wide genetic investigation identifies several SARS-CoV-2-exploited host targets of clinical relevance" for consideration by *eLife*. Your article has been reviewed by 2 peer reviewers, and the evaluation has been overseen by a Reviewing Editor and Jos van der Meer as the Senior Editor. The reviewers have opted to remain anonymous.

Essential Revisions (for the authors):

Please address each of the points raised by both reviewers (8 points for reviewer #1 and 4 points for reviewer #2).

*Reviewer #1 (Recommendations for the authors):*

One of few novel results is the identification of CD209 as a potential alternative SARS-CoV-2 receptor; CD209 is encoded by a gene on chromosome 19 but its concentrations were associated with the lead SNP rs8176719 within the ABO region on chromosome 9, which was strongly linked with several COVID-19 outcomes. This is a good example of a trans-pQTL. The authors mentioned that the interaction of CD209 with SARS-CoV-2 spike protein was recently proposed (but without providing a citation at this point, the citation was provided only much later in the paper). The authors demonstrated an interaction between SARS-CoV-2 spike protein and recombinant CD209 and in cells expressing CD209 on the cell surface. This part is interesting but very under-developed and under-discussed.

1. Specifically, is it suggested that rs8176719 is a pQTL for CD209 through the effect on ABO or through an independent effect? Mediation analysis could be applied to test this hypothesis and a biological mechanism of this association should be discussed. Both of these proteins are measured in soluble form in the blood of controls. How does this relate to the events that occur at the cell surface, presumably during SARS-CoV-2 infection?

2. P.5. Proteins with associated missense variants were flagged and excluded from any further downstream analyses on the basis that the missense variant(s) might influence aptamer binding.

SNP rs8176719 is a protein-affecting frame-shift variant within ABO, why wasn't ABO excluded from the analysis?

3. Table 1: The list of different approaches and their combinations requires a specific dictionary and a more detailed explanation of each approach, this would be a very hard part to understand for the general readership.

4. The statistical considerations: there are clearly many more trans-pQTL interactions tested compared to cis-pQTLs. How was it adjusted for in statistical analyses?

5. P5. Colocalization analysis settings assumed one independent signal in each region. Is this a valid assumption to make and how it might affect the analysis if there is more than one signal?

6. P5. "Genotype data from 10,000 randomly sampled UK Biobank participants was used as a reference panel" – reference panel for what?

7. Supplementary Tables: providing rs numbers in all tables would be very helpful.

8. Zhou et al. (PMID: 33633408) is referenced as both MedRxiv and Nat Med publication.

*Reviewer #2 (Recommendations for the authors):*

1) The authors should cite work by Katz et al. (http://dx.doi.org/10.1101/2020.06.09.20125690), which was the first study to suggest that ABO-mediated change in CD209 levels may play a role in COVID-19 pathogenesis

2) The ABO variation is referred to as rs8176719-TC, which the authors call a SNP. It is not clear what "TC" means and it is not clear why they call the indel a "SNP". It would be helpful if the authors provided more information on this indel variant (which is not a SNP), its consequences on the coding sequence, and linkage to GWAS SNPs.

3) The first phrase in the DC-SIGN section of the Results stating "The ABO signal (rs8176719-TC) overlaps several genes" is unclear and should be clarified.

4) Ref. 10 (medRxiv) is now published in Nature Medicine and the citation should be updated.

---

## [Author Response]

Reviewer #1 (Recommendations for the authors):One of few novel results is the identification of CD209 as a potential alternative SARS-CoV-2 receptor; CD209 is encoded by a gene on chromosome 19 but its concentrations were associated with the lead SNP rs8176719 within the ABO region on chromosome 9, which was strongly linked with several COVID-19 outcomes. This is a good example of a trans-pQTL. The authors mentioned that the interaction of CD209 with SARS-CoV-2 spike protein was recently proposed (but without providing a citation at this point, the citation was provided only much later in the paper). The authors demonstrated an interaction between SARS-CoV-2 spike protein and recombinant CD209 and in cells expressing CD209 on the cell surface. This part is interesting but very under-developed and under-discussed.

We thank the reviewer for their supportive comments and interest in our findings related to CD209. We agree that this is among our more noteworthy novel results, and thus should have been more fully discussed and developed in the text. In our revised manuscript, we have updated each main section (introduction, results, and discussion) to better present these findings. This includes references to the SARS-CoV-2 spike protein and the relevance of its receptors for viral attachment and entry, more description of the experiments we carried out to test the interaction, and an extended discussion that connects our findings with CD209 to other previously-studied viruses that have been studied in the context of both soluble CD209 binding and binding to CD209 as a receptor.

1. Specifically, is it suggested that rs8176719 is a pQTL for CD209 through the effect on ABO or through an independent effect? Mediation analysis could be applied to test this hypothesis and a biological mechanism of this association should be discussed. Both of these proteins are measured in soluble form in the blood of controls. How does this relate to the events that occur at the cell surface, presumably during SARS-CoV-2 infection?

We agree with the reviewer, that a mediation analysis testing whether the effect of the *ABO* signal rs8176719 on COVID risk was mediated by CD209 would be a useful addition to the manuscript. However, as we point out in the paper (and illustrated in Figure 3), the *ABO* signal is highly pleiotropic and is associated with the abundance of several dozen circulating proteins. The pleiotropism of the *ABO* signal (and the limited number of cis-acting independent genetic signals associated with CD209) makes it difficult to tease out the specific contribution of CD209 to COVID risk using the various MR approaches that test genetic mediation (e.g. multivariable or two-step MR methods, which require large numbers of independent genetic signals). Nevertheless, various lines of evidence point to at least partial mediation of the *ABO* signal via CD209: extensive previous literature implicates CD209 as a known viral receptor ^1–5^; the ABO signal shows a stronger association with abundance of soluble CD209 (relative to other *ABO*-signal associated proteins) as noted in our paper; and the binding experiments that we carried out provide evidence of interaction between soluble CD209 and the full-length SARS-CoV-2 spike protein. A complete and formal mediation analysis is warranted in future studies when larger and more well-powered proteomic datasets become available.

With regards to the relationship between soluble and membrane-bound forms of CD209, in our discussion we have also added citations describing the link between the soluble CD209 protein isoform measured in our datasets and the membrane-bound isoform. From these we propose two possible mechanisms: first, because both isoforms correlate in expression, soluble CD209 expression levels could inform the levels of cell-surface CD209 available to facilitate viral entry. Second, the soluble CD209 itself may influence viral infection by competitively binding the virus, analogous to mechanisms previously shown for HIV and cytomegalovirus which both can also directly bind CD209^1,2^.

2. P.5. Proteins with associated missense variants were flagged and excluded from any further downstream analyses on the basis that the missense variant(s) might influence aptamer binding.SNP rs8176719 is a protein-affecting frame-shift variant within ABO, why wasn't ABO excluded from the analysis?

We thank the reviewer for raising this important point that, we believe, requires further clarification on our part. The logic we used is that only missense cis-pQTLs that were not cis-eQTLs would be excluded from analyses. The ABO signal rs8176719, although a frameshift variant, is a cis-eQTL across several eQTL datasets including GTEx and Blueprint (https://genetics.opentargets.org/variant/9_133257521_T_TC, eQTL tab under ‘Assigned genes’ section); this means that causal inference using rs8176719 as a genetic instrument remains valid even though the effect estimates may be biased. We revised parts of the ‘Evidence against aptamer binding artefacts’ section (P6-7) to better reflect this reasoning, mentioned this in the Results section (P9), and flagged ABO in Figure 2 and Figure 5 (where effect estimates were used) with an asterisk denoting that the effect estimates may be biased.

3. Table 1: The list of different approaches and their combinations requires a specific dictionary and a more detailed explanation of each approach, this would be a very hard part to understand for the general readership.

On reflection, we agree with the reviewer, the approaches used in each column of Table 1 require a description in order to be accessible for the general reader. We have provided an extra supplementary table (Supplementary File 9, mentioned in the legend of Table 1) that describes the approaches used in further detail.

4. The statistical considerations: there are clearly many more trans-pQTL interactions tested compared to cis-pQTLs. How was it adjusted for in statistical analyses?

Assuming the statistical considerations raised by the reviewer are related to pan-MR analysis, we would like to direct the reviewer to Supplementary File 2 which shows the number of cis- vs. trans-pQTLs (selected by the GSMR algorithm) used in the pan-MR analysis across 60 proteins that were associated at least one of the seven COVID phenotypes at nominal significance. At nominal significance, there were 1,092 cis-pQTLs vs. 652 trans-pQTLs across 60 protein probes and at 5% FDR, there were 265 cis-pQTLs vs. 240 trans-pQTLs across 9 protein probes.

Overall, our dual pan- and cis-MR approach resulted in 1,286 tests in pan-MR analyses and 2,042 tests in cis-MR analyses (higher than the number of pan-MR tests because of the less stringent p-value used: p <= 1 x 10^-5^) – both these numbers were used to adjust the 5% FDR threshold of significance in pan- and cis-MR analyses, respectively. We have edited the ‘Mendelian randomization’ section (P6) to incorporate this information.

5. P5. Colocalization analysis settings assumed one independent signal in each region. Is this a valid assumption to make and how it might affect the analysis if there is more than one signal?

To keep our analyses simple, we used marginal association statistics to perform genetic colocalisation (clarified further in P6). So yes, we assumed one independent signal in each region. The caveat of this assumption is that we are likely to miss associations which are driven by secondary (weaker) causal variants. We note this limitation in our Discussion (P15):

“However, the genetic colocalisation tests used in our study assumed a single causal variant in each locus and will, therefore, result in higher false negative tests if there is more than one trait-associated causal variant”.6. P5. "Genotype data from 10,000 randomly sampled UK Biobank participants was used as a reference panel" – reference panel for what?

We have edited the ‘Mendelian randomization’ section (P6) to clarify this:

“To select near-independent genetic instruments and account for linkage disequilibrium (LD) in the MR analyses, we used genotype data from 10,000 randomly sampled UK Biobank participants to create a reference LD matrix, which is ancestry-matched to the pQTL data we used.”7. Supplementary Tables: providing rs numbers in all tables would be very helpful.

We used chr:position:ref_allele:alt_allele as the variant identifier because it was a more reliable means of variant identification than rsids. But, as suggested, we have now revised Supplementary File 3, 4, and 8 to include rsids alongside the variant identifiers for ease of look up. The revised supplementary tables should now have an ‘rsid’ column in all tables where the variant IDs (‘snp’ column) are present.

8. Zhou et al. (PMID: 33633408) is referenced as both MedRxiv and Nat Med publication.

We apologise for this duplication. We have now removed all references of Zhou et al. related to the preprint and replaced the reference for the publication (PMID: 33633408)

Reviewer #2 (Recommendations for the authors):1) The authors should cite work by Katz et al. (http://dx.doi.org/10.1101/2020.06.09.20125690), which was the first study to suggest that ABO-mediated change in CD209 levels may play a role in COVID-19 pathogenesis

We agree with the reviewer, Katz et al was the first to suggest a relationship between ABO and CD209 in their preprint. We would also like to note that, after the preprint, both Katz et al and ourselves simultaneously published this discovery as a Correspondence in NEJM (https://www.nejm.org/doi/full/10.1056/NEJMc2025747) and it is this publication that was cited in our paper. The correspondence from Katz et al included the text from their preprint (https://www.nejm.org/doi/suppl/10.1056/NEJMc2025747/suppl_file/nejmc2025747_sa2_appendix.pdf). Nevertheless, we have now mentioned the preprint in the Discussion section (P13).

2) The ABO variation is referred to as rs8176719-TC, which the authors call a SNP. It is not clear what "TC" means and it is not clear why they call the indel a "SNP". It would be helpful if the authors provided more information on this indel variant (which is not a SNP), its consequences on the coding sequence, and linkage to GWAS SNPs.

We regret this confusion. In the revised manuscript, we have edited rs8176719-TC to be rs8176719-insC (or insertionC) to make it explicit what the effect allele is and describe the variant and clarify the consequences of the variant on P9 when it is first mentioned. We have also re-labelled rs8176719 as an ABO signal rather than a ‘SNP’.

Furthermore, previously we used LDlink to annotate the functional consequences of variants and in many cases, including for the ABO variant, we discovered that the variants were not properly annotated. In this revised version, we use gnomad v2 variant effect prediction annotations from our portal Open Targets Genetics to annotate the functional consequence of all variants (revised Supplementary File 3 and 4).

3) The first phrase in the DC-SIGN section of the Results stating "The ABO signal (rs8176719-TC) overlaps several genes" is unclear and should be clarified.

We edited the text in that section (P9) and replaced "The ABO signal (rs8176719-TC) overlaps several genes" with “The ABO signal (rs8176719-insC) contributes to the determination of non-O blood groups and regulates circulating levels of both ABO and several non-ABO proteins”.

4) Ref. 10 (medRxiv) is now published in Nature Medicine and the citation should be updated.

As noted in Comment #8 from Reviewer #1, we have updated Zhou et al’s references to include only the Nature Medicine publication.

References

1. Geijtenbeek, T. B. *et al.* DC-SIGN, a dendritic cell-specific HIV-1-binding protein that enhances trans-infection of T cells. *Cell* 100, (2000).

2. Plazolles, N. *et al.* Pivotal advance: The promotion of soluble DC-SIGN release by inflammatory signals and its enhancement of cytomegalovirus-mediated cis-infection of myeloid dendritic cells. *J. Leukoc. Biol.* 89, (2011).

3. Yang, Z.-Y. *et al.* pH-dependent entry of severe acute respiratory syndrome coronavirus is mediated by the spike glycoprotein and enhanced by dendritic cell transfer through DC-SIGN. *J. Virol.* 78, 5642–5650 (2004).

4. Londrigan, S. L. *et al.* N-Linked Glycosylation Facilitates Sialic Acid-Independent Attachment and Entry of Influenza A Viruses into Cells Expressing DC-SIGN or L-SIGN. *Journal of Virology* vol. 85 2990–3000 (2011).

5. Johnson, T. R., McLellan, J. S. & Graham, B. S. Respiratory syncytial virus glycoprotein G interacts with DC-SIGN and L-SIGN to activate ERK1 and ERK2. *J. Virol.* 86, 1339–1347 (2012).